# Characterization of BRCA Deficiency in Ovarian Cancer

**DOI:** 10.3390/cancers15051530

**Published:** 2023-02-28

**Authors:** Giovanna Barbero, Roberta Zuntini, Pamela Magini, Laura Desiderio, Michela Bonaguro, Anna Myriam Perrone, Daniela Rubino, Mina Grippa, Antonio De Leo, Claudio Ceccarelli, Lea Godino, Sara Miccoli, Simona Ferrari, Donatella Santini, Pierandrea De Iaco, Claudio Zamagni, Giovanni Innella, Daniela Turchetti

**Affiliations:** 1Medical Genetics Unit, IRCCS Azienda Ospedaliero—Universitaria di Bologna, 40138 Bologna, Italy; 2Department of Medical and Surgical Sciences, University of Bologna, 40138 Bologna, Italy

**Keywords:** *BRCA1*, *BRCA2*, ovarian cancer

## Abstract

**Simple Summary:**

Ovarian cancer (OC) is a highly lethal malignancy. Major improvements in treatment are expected from the identification of molecular features that may predict outcome or be used as therapeutic targets. Among genetic defects relevant for OC are those of *BRCA1* and *BRCA2* genes. Indeed, at least 20% of OC patients carry inherited or acquired *BRCA1/2* pathogenic variants, the identification of which is important for treatment and prevention. A comprehensive study of 30 OC patients revealed that 7 (23%) had BRCA alterations (6 inherited and 1 acquired) detectable by usual clinical testing, while another 5 patients (17%) showed epigenetic silencing of *BRCA1* in the tumor, which would have escaped standard sequencing analysis, and one had an inherited variant in another gene: *RAD51C*, involved in the same DNA repair mechanism as *BRCA1* and *BRCA2*. Patients with BRCA deficit showed greater genomic instability, but better survival, than those with no evidence of BRCA deficit.

**Abstract:**

BRCA testing is recommended in all Ovarian Cancer (OC) patients, but the optimal approach is debated. The landscape of *BRCA* alterations was explored in 30 consecutive OC patients: 6 (20.0%) carried germline pathogenic variants, 1 (3.3%) a somatic mutation of *BRCA2*, 2 (6.7%) unclassified germline variants in *BRCA1*, and 5 (16.7%) hypermethylation of the *BRCA1* promoter. Overall, 12 patients (40.0%) showed BRCA deficit (BD), due to inactivation of both alleles of either *BRCA1* or *BRCA2*, while 18 (60.0%) had undetected/unclear BRCA deficit (BU). Regarding sequence changes, analysis performed on Formalin-Fixed-Paraffin-Embedded tissue through a validated diagnostic protocol showed 100% accuracy, compared with 96.3% for Snap-Frozen tissue and 77.8% for the pre-diagnostic Formalin-Fixed-Paraffin-Embedded protocol. BD tumors, compared to BU, showed a significantly higher rate of small genomic rearrangements. After a median follow-up of 60.3 months, the mean PFS was 54.9 ± 27.2 months in BD patients and 34.6 ± 26.7 months in BU patients (*p* = 0.055). The analysis of other cancer genes in BU patients identified a carrier of a pathogenic germline variant in *RAD51C*. Thus, BRCA sequencing alone may miss tumors potentially responsive to specific treatments (due to *BRCA1* promoter methylation or mutations in other genes) while unvalidated FFPE approaches may yield false-positive results.

## 1. Introduction

Ovarian cancer (OC) is the most lethal gynecological neoplasm, with an average overall survival of about 40% at 5 years from diagnosis [1,2]. The search for molecular defects which can affect disease outcomes and constitute therapeutic targets is therefore a priority to improve the management of OC patients. Among those, *BRCA1/2* germline variants have been reported in about 14% of cases [3], while the fraction of OC with somatic *BRCA* mutations is generally reported to be between 3% and 9% [4,5].

In particular, several studies have shown that germline or somatic *BRCA1/2* pathogenic variants predict greater sensitivity to standard platinum- and taxane-based therapies [6,7,8] and to maintenance treatments with Poly (ADP-ribose) Polymerase (PARP) inhibitors [9]. The therapeutic efficacy of the latter, which intervene in single-stranded DNA repair, is achieved through a mechanism of “synthetic lethality” in the presence of a concomitant loss of function of the double-stranded DNA repair mechanisms by homologous recombination (HR), in which BRCA1/2 proteins play an essential role [4,10,11,12].

BRCA genetic testing usually implies sequencing the coding portion and searching for deletion/duplications of the *BRCA1/2* genes [13,14,15]. The traditional approach relies on the analysis of DNA extracted from the peripheral blood of the patients, which allows the detection of “constitutional” or “germline” variants. Recently, the evidence that about 1/3 of *BRCA1/2* pathogenic variants in OC patients are confined to the tumor tissue [4,10], has led to a recommendation that BRCA analysis be performed on DNA extracted from cancer tissue, in order to detect both the constitutional and the somatic variants [16].

However, cancer tissue testing poses some critical issues, such as: differences between the types of samples, including formalin-fixed and paraffin-embedded (FFPE) tissues and snap-frozen (SF) tissues, the choice between primary tumor and relapse, the assessment of large rearrangements and the predictive value of specific variants for drug response [17,18,19,20]. Furthermore, a non-negligible fraction of ovarian tumors (11–16%) present BRCA deficiency due to epigenetic inactivation of *BRCA1*, not identifiable with routine somatic tests, and some tumors may present homologous recombination deficiency (HRD) due to alterations in other genes of the pathway [5,21,22,23].

In this work, we have performed a comprehensive assessment of BRCA defects in tissues from 30 clinically characterized OC patients in order to explore the landscape of genetic alterations and evaluate the accuracy of standard diagnostic testing.

The primary aim of the study was to characterize OC samples of newly diagnosed patients for the presence of mutations, rearrangements, or epimutations of the *BRCA1/2* genes and to validate tissue testing strategies. Secondary aims were to further dissect the molecular features of the samples, by assessing genomic rearrangements in BRCA-defective tumors and mutations in genes other than *BRCA1/2* in tumors with no BRCA defects detected, and to assess clinical outcome according to BRCA status.

## 2. Materials and Methods

### 2.1. Patients, Clinical Data, and Tumor Specimens

The GeCO (Genetic Characterization of Ovarian cancer) study protocol was approved by the Ethical Board of S.Orsola-Malpighi Hospital, Bologna, Italy (Prot. 81/2014/U/Tess) and was conforming to the ethical guidelines of the WMA Declaration of Helsinki.

Patients were considered eligible for the study if the following inclusion criteria were fulfilled:newly diagnosed OC;major age;informed consent.The exclusion criteria were:borderline, stromal, and/or mucinous type OCs;unavailability of tumor tissue samples suitable for molecular analysis.

Thirty-nine consecutive newly diagnosed OC patients admitted to the Gynecological Oncology unit of S.Orsola-Malpighi Hospital in the first semester of 2015 to undergo surgical procedures were proposed for the study and 38 were accepted to be enrolled. Before surgery, patients underwent a genetic counseling session during which, after accurate collection of family history, they were informed in detail about the aims and implications of the study. Upon informed consent, a venous blood sample was drawn; then, immediately after surgery, tumor tissue was dissected by the pathologist, and a sample was snap-frozen.

After the exclusion of 8 patients (5 because the histologic types were different from those eligible and 3 because tissue samples were not adequate for the analysis), 30 were included in the study.

For included patients, 10 µm slides of FFPE tissue with a percentage of tumor cells greater than 70% were also prepared for genetic analysis. Clinicopathological data, including age at diagnosis, tumor location, histologic type, grade, stage, and type of surgery and therapy were collected from medical records and pathology reports. Follow-up data were updated on a regular basis until December 2022 by checking on clinical charts the situation of each patient at their last access to the Oncology Unit.

### 2.2. Nucleic Acid Isolation

DNA was extracted from peripheral blood, frozen tissue, and (FFPE) tissue using QIAamp DNA Mini Kit (Qiagen, Hilden, Germany) according to the manufacturer’s instructions.

RNA was isolated from frozen tissue stabilized in RNAlater (Qiagen, Hilden, Germany) using Rneasy Mini Kit (Qiagen, Hilden, Germany) according to the manufacturer’s instructions. DNase treatment was performed using an RNase-Free DNase set (Qiagen, Hilden, Germany). DNA and RNA were quantified using a Nanodrop spectrophotometer (Thermo Fisher Scientific, Waltham, MA, USA).

### 2.3. Genomic BRCA1 and BRCA2 Analysis

For the first 23 patients, the analysis of DNA extracted from SF tumor was performed using either Sanger sequencing or next-generation sequencing (NGS), to allow comparison between the two sequencing methods and increase accuracy, while germline DNA analysis was carried out through NGS only. Sanger sequencing was performed on coding exons and splice site junction of *BRCA1* and *BRCA2* genes (NM_007294.3 and NM_000059.3 respectively) using “BigDye Terminator v1.1 Cycle Sequencing Kit” and analyzed on an automatic genetic sequencer (ABI3730 DNA Analyzer, Thermofisher); NGS analysis was performed using an Ion AmpliSeq BRCA1/2 Panel (Thermo Fisher Scientific, Waltham, MA, USA) under standard conditions. Briefly, 30 ng of DNA was used to set manually libraries with Ion AmpliSeq Library Kit v.2.0 and IonXpress Barcode Adapter Kit. A template was prepared with Ion PGM TM 510TM & 520 TM & 530 TM kit—Chef using the Ion OneTouch 2 InstrumentChef System. Sequencing was performed on an Ion PGM System using Ion 318 chip and Ion PGM Sequencing 200 Kit v2. NGS data were analyzed with Torrent suite and Ion Reporter Software, version 5.6 and later.

For the last seven patients, both SF tumor tissue and constitutional DNA were analyzed by NGS using Oncomine BRCA Research Assay (Thermo Fisher Scientific, Waltham, MA, USA), made available at our center in the meantime, under standard conditions. Briefly, 20 ng of DNA was used to prepare manually libraries with an Ion AmpliSeq Library Kit Plus and IonXpress Barcode Adapter Kit. A template was prepared with Ion 520 & 530 Kit OT2 using an Ion OneTouch 2 Instrument and an Ion OneTouch ES Instrument. Sequencing was performed on an Ion S5 System using Ion 520 chip. NGS data were analyzed with Torrent suite and Ion Reporter Software 5.10.

DNA extracted from all FFPE tumor samples was analyzed using Oncomine BRCA Research Assay (Thermo Fisher Scientific, Waltham, MA, USA) as described above. In more detail, in the first assay (Research FFPE, 2017), 20 ng of DNA, extracted from not-deparaffinized FFPE samples, was used to prepare manually libraries with Ion AmpliSeq Library Kit Plus and IonXpress Barcode Adapter Kit. A template was prepared with Ion PGM TM 510TM & 520 TM & 530 TM kit—Chef using the Ion OneTouch ES InstrumentChef System. Sequencing was performed on an Ion PGM System using Ion 318 chip and Ion PGM Sequencing 200 Kit v2. NGS data were analyzed with Torrent suite and Ion Reporter Software version 5.6 and later. In the second analysis (Diagnostic FFPE, 2020) 20 ng of DNA, extracted from deparaffinized FFPE samples, were used to prepare Chef-Ready libraries with Ion AmpliSeq TM Kit for Chef DL8 and IonXpress Barcode Adapter Kit. A template was prepared with Ion 510 TM & 520 TM & 530 TM Kit—Chef using an Ion Chef TM Instrument. Sequencing was performed on an Ion S5 System using Ion 520 chip. NGS data were analyzed with Torrent suite and Ion Reporter Software version 5.10 and later. Targeted sanger sequencing was performed to check C3 (VUS), C4 (likely pathogenic), and C5 (pathogenic) variants in respective constitutional DNA.

Deletion and duplication of *BRCA1/2* genes were analyzed in frozen tissue and blood samples using MLPA techniques (P002-D1 *BRCA1* and P045-C1 *BRCA2/CHEK2*—MRC-Holland, Amsterdam, the Netherlands) under the manufacturer’s protocol. Fragments were separated on an ABI3730 DNA Analyzer and analyzed with Coffalyzer.net Software.

### 2.4. Methylation-Specific MLPA (MS-MLPA)

Methylation analysis of the *BRCA1/2*-gene promoter was performed using MLPA ME053 probemix kit (MRC-Holland, Amsterdam, The Netherlands). This kit contains specific probes for CpG islands: three in the *BRCA1* gene and four in the *BRCA2* gene. In addition, there are four probes for copy number variation (CNV) detection of the *BRCA1* gene (targeting exons 3, 13, 20, 23) and four probes for CNV detection of the *BRCA2* gene (targeting exons 3-13-17-21). Fragments were separated on an ABI3730 DNA Analyzer and analyzed with Coffalyzer.net Software.

### 2.5. Heterozygosity Analysis

Heterozygosity status was assessed through the analysis of 16 microsatellites mapping on chromosomes 17 and 13 (panels 23, 24, and 19 respectively; Thermo Fisher Scientific, Waltham, MA, USA). For chromosome 17 we selected 11 markers: D17S849, D17S831, D17S938, D17S1852, D17S799, D17S798, D17S1868, D17S949, D17S785, D17S784, D17S928; and five markers for chromosome 13: D13S171, D13S153, D13S265, D13S159, D13S158. Polimeration chain reaction (PCR) was performed using Kapa Taq HotStart DNA Polymerase (KAPA Biosystems, Wilmington, MA, USA) under standard conditions and run on an ABI3730 DNA Analyzer. Data were analyzed using GeneMapper Software (Thermo Fisher Scientific, Waltham, MA, USA).

### 2.6. Gene Expression Analysis by Droplet Digital PCR (ddPCR)

Reverse transcription was performed using 500 ng of RNA using iScript Reverse Transcription Supermix for RT-qPCR (BioRad Laboratories, Hercules, CA, USA). Two multiplex reactions were performed including both target and reference genes and using validated assays for *BRCA1* (qHsaCEP0041326), *BRCA2* (qHsaCEP0052184) (FAM probes), and reference gene PPIA (qHsa CEP0041342) (Hex probe) (BioRad Laboratories, Hercules, CA, USA). Briefly, PCR reactions were conducted using 1 ng of cDNA and ddPCR Supermix for Probes according to the manufacturer’s instructions. Droplets were generated by loading reaction mixtures and Droplet Generation Oil for Probes into a DG8 Cartridge using a QX200 Droplet Generator (BioRad Laboratories, Hercules, CA, USA). Samples were carefully transferred in-plate, sealed, and run on a thermocycler. Finally, the plates were transferred in the QX200 Droplet Reader and data were acquired and analyzed using QuantaSoft software.

### 2.7. NGS Analysis of Other Cancer Genes

Tumor samples with no evidence of BRCA deficiency were subjected to sequencing of other candidate genes in order to identify any different molecular mechanisms underlying carcinogenesis. To this aim, a custom Ion AmpliSeq On-Demand panel (Thermo Fisher Scientific, Waltham, MA, USA) was used, designed to detect SNV and small indel variants in 21 genes associated with cancer predisposition: *APC*, *ATM*, *BMPR1A*, *BRIP1*, *CHEK2*, *EPCAM*, *MLH1*, *MSH2*, *MSH3*, *MSH6*, *MUTYH*, *PALB2*, *RAD51C*, *RAD51D*, *PTEN*, *PMS2*, *POLD1*, *POLE*, *SMAD4*, *STK11*, *TP53*. DNA analysis of the FFPE tumor samples was performed under standard conditions. Briefly, 20 ng of DNA was used to prepare manually libraries with Ion AmpliSeq Library Kit Plus and IonXpress Barcode Adapter Kit. A template was prepared with 510TM &520TM &530TM kit Chef using the Chef System. Sequencing was performed on an Ion S5 System using Ion 530 chip. NGS data were analyzed with Torrent suite and Ion Reporter Software 5.16.

Class 4 or 5 variants (according to ClinVar classification (https://www.ncbi.nlm.nih.gov/clinvar/ accessed on 12 December 2022) in genes other than *TP53* (which is expected to be somatically mutated in a substantial proportion of ovarian carcinomas) were searched for in the patient’s blood sample.

### 2.8. Array-CGH+SNP Analysis

For 13 SF DNA samples, CNV and LOH analysis was performed using GenetiSure Cancer Research CGH+SNP Microarray, 2 × 400 K (Agilent Technologies, Santa Clara, CA, USA), according to the manufacturer’s protocol, with appropriate Agilent reference DNAs (Euro female). The microarray contains approximately 300,000 in situ synthesized 60-mer oligonucleotides with a medium resolution of 30 kb (higher resolution in cancer-associated genes) and 103,000 SNP probes. The array data extraction and analysis were performed using CytoGenomics v.5.2 (Agilent Technologies, Santa Clara, CA, USA). Aberrations were detected using the ADM-2 algorithm with a threshold of 6.0.

Due to the low quality of the DNA samples, some modifications were made to the protocol to improve the quality of the experiment and subsequent analysis:

(1) a dye-swap design was used. DNA samples were labeled with cyanine 3 which has greater stability than cyanine 5;

(2) different amounts of DNA for samples and reference were used in order to obtain a better yield and increase specific activity. Digestion, labeling, and hybridization were performed using 1500 ng of test DNA and 1000 ng of reference DNA.

The analysis was performed simultaneously for CNVs and LOH detection. Only CNVs larger than 1 Mb and with a threshold of log2ratio > 0.2 for gain and <−0.2 for loss were considered.

CNVs were initially classified based on the type of aberration (copy loss and copy gain) and then divided into “simple” and “complex” loss or gains. “Simple” CNVs were defined by a single aberrant mean log2ratio, while “complex” CNVs were split in two or more regions with different log2ratios, possibly indicating distinct cellular clones with differences in CNV length and/or copy number in that chromosomal location.

CNV features (number and average size) were compared between BRCA defective and intact patients and their distributions were compared through the Kolmogorov-Smirnov test.

### 2.9. Statistical Analysis

The clinical-pathological data were organized into nominal variables and were analyzed using the “Statistical Package for Social Science (SPSS)” software, version 25.0 (SPSS, Chicago, IL, USA). Two-tailed *p*-values less than 0.05 were considered statistically significant.

Mean, standard deviation (SD), ranges, and frequencies were used as descriptive statistics. Progression-free Survival (PFS) is defined as the elapsed time between the date of initial diagnosis and either the date of recurrence or the last follow-up. Overall Survival (OS) is defined as an estimate from the date of initial diagnosis to the date of death or the last follow-up (if death was not observed during the follow-up period). OS and PFS were estimated using the Kaplan–Meier method with STATA software, version 13.0. A Cox regression model was used to estimate the hazard ratio and its 95% CI. Follow-up times were described as medians.

## 3. Results

### 3.1. BRCA1/2 Sequence Variants

*BRCA1/2* sequence analysis on SF OC tissues identified seven (23.3%) pathogenic variants (three in *BRCA1* and four in *BRCA2*) and three (10.0%) variants of uncertain significance (two in *BRCA1* and one in *BRCA2)*. Among *BRCA2* variants, p.Asn1784Lys (C3) and p.Ser2148Leufs*20 (C5) presented with an allele load consistent with the heterozygous status and were found to be exclusively somatic after a targeted search in peripheral blood. Conversely, all the six pathogenic variants that were subsequently found to be germline had a frequency consistent with the homozygous status in tumor tissue (VAF: 80–100%). The variants detected in *BRCA* genes are reported in Table 1.

Variants were reported according to HGVS nomenclature using as reference sequence NM_007294.3 for *BRCA1* and NM_000059.3 for *BRCA2*.

### 3.2. BRCA Copy Number Variants and Loss of Heterozygosity

Rearrangements revealed by MLPA affected *BRCA1* in 23 (87%) samples and *BRCA2* in 14 (57%); among those, 20 *BRCA1* and 14 *BRCA2* rearrangements involved the deletion/duplication of the whole allele, while partial *BRCA1* rearrangements were observed in GECO 15 (deletion from exon 1 to exon 11), GECO 22 (duplication from exon 11 to the end) and GECO 31 (duplication of exons 1 and 2).

Microsatellite analysis showed loss of heterozygosity (LOH) at both *BRCA1* and *BRCA2* regions in 16 (53%) samples. All six patients carrying germline pathogenic variants displayed LOH; this was caused by the deletion of the wild-type allele in four cases; of the other two, GECO 29 showed a copy neutral LOH (CN-LOH), GECO 31 a partial duplication associated with LOH of the entire chromosome. Instead, samples of the two patients carrying germline variants of uncertain significance in *BRCA1* displayed the loss of the allele harboring the variant in the tumor, due to allele deletion (GECO 2) or to CN-LOH (GECO 27). Moreover, microsatellite analysis showed that in three cases with duplication of the *BRCA2* gene, the entire chromosome was duplicated (Examples of microsatellite analysis are shown in Appendix A).

### 3.3. Methylation and Gene Expression Results

MS-MLPA analysis performed on tumor tissue samples showed that *BRCA1* promoter hypermethylation was present in five samples (17%), while no samples showed*BRCA2* promoter hypermethylation. In four cases, *BRCA1* promoter hypermethylation co-existed with a somatic *BRCA1* deletion, while in the remaining case, both alleles presented hypermethylation, and CN-LOH was shown (Appendix A).

*BRCA1/2* gene expression was evaluated by Digital PCR in 24 OC samples (for the remaining six, RNA was inadequate for the analysis). Gene expression levels were defined as increased, reduced, or normal by comparing gene expression in each tumor sample with alterations to gene expression levels in samples without gene alterations: variations greater than two-fold SD were considered reliable variations. *BRCA1* expression was shown as decreased in three samples with *BRCA1* promoter hypermethylation (GECO 3, GECO 5, and GECO 34), and in one sample harboring a pathogenic variant (GECO 31). Two samples (GECO 24 and GECO 30) showed a reduction in both *BRCA1* and *BRCA2* gene expression, which was associated with LOH in both genes and, in GECO 24, with a pathogenic variant in *BRCA2*. Gene expression increased in three samples, two with *BRCA1* (GECO 14 and GECO 23), and one with *BRCA2* increase (GECO 15).

### 3.4. Classification of BRCA Deficit

Tumors showing evidence of structural or functional loss of both the alleles of either *BRCA1* or *BRCA2* (for carrying a pathogenic germline or somatic *BRCA1/2* variant and lacking the wild-type allele, or presenting with deletion of one *BRCA1/2* copy and promoter methylation of the other), were classified as “BRCA-deficient” (BD), while tumors in which evidence of BRCA deficit was absent or inconclusive were defined as “BRCA deficit undetected/unclear” (BU).

Overall, twelve patient samples (40%) were classified as BD: four (33.3%) because of germline pathogenic variant of one allele and partial/complete deletion of the other (one *BRCA1* and three *BRCA2*), four (33.3%) because of partial/complete *BRCA1* deletion and promoter methylation of the other allele, two (16.7%) because of a pathogenic variant of one allele (one *BRCA1* germline variant and one *BRCA2* somatic variant) and CN-LOH, one (8.3%) because of germline *BRCA1* pathogenic variant and *BRCA1* partial duplication and one (8.3%) because of promoter methylation of both *BRCA1* alleles (CN-LOH was present). These results are summarized in Figure 1.

### 3.5. Cancer Genes Panel Results

The NGS analysis of a multigene panel of other cancer-predisposing genes was performed on tumor samples of the 18 BU patients and detected *TP53* mutations in 7 samples (38.9%).

In addition, two C4/C5 variants were detected in two patients: *RAD51C* c.904 + 5G > T was found in patient GECO 14 and was shown to be germline, while *PTEN* c.388C > T;p.Arg130Ter, found in GECO 22, was excluded in the germline.

Panel results are detailed in Appendix A.

### 3.6. Array-CGH + SNP Analysis Results

Array-CGH + SNP analysis was performed to identify CNVs and LOH in six BU and seven BD patients. Table 2 summarizes the main results and shows the comparison between BD and BU sample sets, with p-values from a Kolmogorov–Smirnov test.

The number and average size of global CNVs and separately of duplications and deletions were evaluated, and further divided into “simple” and “complex” from array-CGH profiles. The analysis revealed a great complexity of unbalanced chromosomal rearrangements in OC samples with about half of the genome involved (Table 2), as expected for high-grade cancers. The identification of CNVs composed of multiple segments with different log2ratios further supported the chromosomal heterogeneity of the analyzed samples.

Statistically significant differences emerged in the number of total CNVs and duplications, especially “simple” gains, which are more numerous in BD samples. Deletions tended to be larger in BU samples, although without statistical significance. Interestingly, BD GECO 7 and 27, with less advanced OC (IIb and Ic, respectively) are the patients with the highest number of CNVs, showing that the chromosomal picture has evolved more rapidly and earlier than the biological features of the tumoral tissues, probably fostered by the deficiency of BRCA-related repair mechanisms. Conversely, GECO 18 has a very preserved genome despite its advanced stage (IIIc).

### 3.7. FFPE Analyses

NGS-based BRCA analysis of FFPE samples from 29 patients was performed in order to assess whether the results obtained on this type of sample were consistent with those found in SF from the same surgical specimen. Sequencing analysis provided results satisfying quality assessment in 27 cases (on target > 85%, Mean depth > 500, Uniformity > 85%), while two cases (GECO 16 and GECO 27) did not present with adequate quality.

FFPE analysis was first carried out in 2017 to assess the accuracy of BRCA analysis on FFPE, according to the study design (Research FFPE analysis: “R-FFPE”); all the variants identified on SF samples were confirmed except one (GECO 3): the absence was confirmed in a different FFPE block from the same surgery. However, additional mutations were found in 24 out of 27 samples analyzed (88.9%); particularly, setting the allelic load threshold at 5%, C4–C5 mutations were retrieved in 14 samples with no pathogenic/likely pathogenic variants previously detected. Setting the threshold at 20%, 33 additional C3–5 variants were found in nine patients, as reported in Table 3.

The analysis was repeated in 2020 (excluding samples with clear pathogenic variants and those with no additional mutations detected in tumor tissue), when the diagnostic analysis of FFPE had been implemented and used for three years in the clinical setting (Diagnostic FFPE analysis: “D-FFPE”); all the variants identified by testing SF tissues were confirmed with the exception, again, of the GECO 3 variant, but, unlike in R-FFPE, no additional variants were detected, suggesting that the additional mutations detected by R-FFPE were false findings. The comparison between the results of the sequence analyses performed on SF tissues and FFPE tissues at the two time points is shown in Table 3.

Assuming as true the findings replicated in at least two assays and considering as a positive result the presence of at least one C3–C5 variant in a sample and as a negative result the absence of any variant, sensitivity was estimated to be 100% for all the approaches (SF, R-FFPE, and D-FFPE), while specificity was 95% for SF, 70% for R-FFPE and 100% for D-FFPE, with an accuracy of 96.3%, 77.8%, and 100%, respectively.

### 3.8. Clinical Characterization and Correlations with BRCA Status

The clinical characteristics and outcomes of the 30 OC patients enrolled in the study are summarized in Table 4.

Patients were subdivided into two groups based on the presence (BD) or absence (BU) of BRCA deficiency revealed by the analysis performed on tumor tissue samples. As shown in Table 5, baseline clinical characteristics of the two groups were similar: median age at diagnosis was 50.8 (±12.9) years in the BD group and 58.9 (±7.5) years in the BU group (*p* = 0.070); the majority of tumors were high-grade papillary-serous carcinomas (only one case of endometrioid carcinoma in BD group); 10 patients of the BD group (83.3%) and 15 of the BU group (88.2%) presented with advanced FIGO (International Federation of Gynecology and Obstetrics stages III and IV) stage disease (*p* = 1.000); considering that the type of surgery performed was the same in both groups (Hysterectomy with Bilateral Salpingo-Oophorectomy), four patients in the BD group (33.3%) and one in the BU group (5.9%) had macroresidual post-surgery (R) > 0, (*p* = 0.130). Most patients (22) underwent adjuvant chemotherapy with carboplatin and paclitaxel, three with only carboplatin, and one patient did not undergo chemotherapy because of poor clinical conditions at diagnosis.

After a median follow-up of 60.5 months, 18 (60.0%) patients had relapsed and 10 (33.3%) had died. For each patient, interval to disease progression and interval to death (in months) were evaluated to reveal any differences in PFS and OS between patients with BRCA deficiency and patients without deficit (GECO 2 was excluded from PFS calculation because she was never free from disease and died a few months after diagnosis): as reported in Table 5 and Figure 2, mean PFS was 54.9 ± 27.2 months in BD patients and 34.6 ± 26.7 months in BU patients (*p* = 0.055), while mean OS was 69.1 ± 20.0 months in BD patients and 58.4 ± 23.5 months in BU patients (*p* = 0.077).

## 4. Discussion

Impairment of BRCA function in OC has proven to predict response to platinum-based chemotherapy and PARP-inhibitors; consequently, *BRCA1/2* analysis is being routinely used to inform the medical treatment of OC patients [24,25,26]; the advantage of identifying also somatic *BRCA1/2* mutations has led to the recommendation that *BRCA* sequencing be performed on tumor tissue. However, heterogeneity in diagnostic approaches and result interpretation raises uncertainties regarding the clinical meaning of somatic findings [16,17].

To contribute to elucidating the landscape of BRCA defects and evaluating the ability of clinical testing to correctly identify them, we extensively analyzed BRCA alterations in a consecutive series of 30 well-characterized OCs. Consistently with previous evidence [27,28,29,30], we found that a substantial fraction of OCs (40.0%) presented with BRCA deficit, here defined by the presence of alterations predicted to result in the complete absence of functional copies of either *BRCA1* or *BRCA2* gene, due to sequence variants, rearrangements or epigenetic silencing.

As only half of BD cases harbored germline *BRCA1/2* variants, our results support the superiority of testing approaches involving tumor tissue analysis in detecting potentially actionable alterations. However, 16.7% of samples (41.7% of those classified as BD) displayed *BRCA1* promoter hypermethylation, which would be missed by standard somatic tests that are based on gene sequencing [4,5,24,31]. Conversely, it has been suggested that promoter hypermethylation, if compared to gene mutations, may be more easily removed under the selective pressure induced by treatment, leading to a higher chance of drug resistance development [32]. All the samples with *BRCA1* promoter hypermethylation showed LOH at the BRCA1 locus, suggesting the absence of unmethylated alleles, and were therefore classified as BD. Gene-expression analysis, however, failed to show a reduction in two out of three methylated samples, as it did in four of six samples with germline pathogenic variants associated with LOH in cancer. Although it is possible that an allele carrying a pathogenic variant is expressed, taken together, these findings suggest that the expression assay was not able to provide reliable information on BRCA deficiency.

After assessing BRCA status through such a combined approach, we aimed at exploring the performance of FFPE-based BRCA sequencing, that is the BRCA test mainly used for predictive purposes in OC, in correctly classifying BRCA-deficient tumors. The first analysis, carried out in 2017 for research purposes, showed a plethora of additional mutations, including potentially significant (C3–5, allelic load > 20%) variants in nine patients if compared to SF analysis. In a total of 14 patients the detection of C4/C5 mutations (allele load > 20% in two, 5–20% in 12) would have changed the treatment based on current practice. A second analysis, performed in 2020 according to diagnostic standards, did not confirm those findings, since no variants were detected in addition to those found in SF samples, with the specificity raising from 70% to 100%. This increased accuracy can be explained by technical improvements made in the analytical approach before moving to diagnostic routine, which included prior de-paraffinization of samples and instrumental upgrades. Nevertheless, the high rate of false-positive results in the first analysis should alert about the reliability of somatic testing performed by inexperienced laboratories, and underlines the need for proper validation and adherence to verified protocols and quality controls [16,17,33]. In any case, the variant load in tissue appears to be a crucial issue for clinical interpretation. Indeed, all the validated variants predicted to lead to BRCA deficiency showed an allelic load in tumor tissue of 50% or higher. Regarding these as “predictive” mutations with a frequency lower than 20% in tumor DNA (provided that the proportion of tumor cells in the sample is adequate) may pose serious risks of misinterpretation, with implications for therapeutic choices. First, the lower the allele frequency, the higher the chance of a false positive result due to artifacts, as suggested by our findings; second, even if true, a low-load alteration is consistent with a normal copy of the gene being retained, leading to BRCA proficiency in the cell, that is the reason why we made the conservative choice to regard as BRCA-deficient only samples with evidence of no functional copies of either the *BRCA1* or *BRCA2* gene. Interestingly, two germline C3 variants were found at low frequency in tumor DNA, suggesting the loss of the allele harboring the variant in cancer cells: such finding provides evidence against pathogenicity that can eventually contribute to variant classification and supports the usefulness of combining germline and somatic testing.

Another C3 variant was found in SF tissue, though not confirmed in the other samples from the same patients; as artifacts are less common in SF tissue, it can be hypothesized that the mutation occurred in a subpopulation of tumor cells, which underlines that tumor heterogeneity and the chance of passenger mutations should be taken into account when interpreting somatic test results for clinical purpose.

In addition to gene mutations, CNVs were found in the majority of samples, confirming the frequency of high genomic instability in OC, irrespective of BRCA status. In patients with pathogenic BRCA variants or epigenetic silencing, rearrangements at BRCA loci accounted for LOH, according to the expected two-hit mechanism; however, in other patients, CNV at BRCA loci were not associated with alterations of the other allele, and could be viewed as an aspecific manifestation of genomic instability, not supporting the usefulness of including the analysis of somatic CNV in predictive BRCA testing.

Indeed, genomic scars, including CNVs and LOH, are signs of genomic instability due to HRD. SNP arrays and more recently NGS have been used to detect these chromosomal anomalies to calculate an HRD score, predicting patients that might be responsive to PARP inhibitors [34,35]. Genomic rearrangements were assessed in a subgroup of samples in order to compare BRCA-deficient with non-deficient tumors. The analysis of deletions and duplications did not lead to statistically significant differences between BRCA intact and defective cancers; as already reported for serous OC [36], highly rearranged genomes were found, both in BRCA intact and defective cancers, with a complex heterogeneity of cellular clones with different chromosomal anomalies that cannot be fully appreciated and precisely defined by array-CGH. However, when the analysis was extended to small CNVs (>1 Mb and <10 Mb), the frequency, especially of duplications, was significantly higher in OC with BRCA deficiency, which is consistent with a previous report [37]. Considering the total number of CNVs and the average size of deletions, GECO 6 showed values more similar to those of patients presenting BRCA deficiency, suggesting a possible HRD, that, however, was not explained by sequencing other cancer genes.

As for clinical outcome, although there were no significant differences by baseline prognostic factors and by treatment, BD, if compared to BU, patients showed a tendency to a better survival (not reaching, however, statistical significance); since at the time of the study, PARP inhibitors were not used as maintenance treatment after first-line chemotherapy, the prolonged PFS is likely the result of a better response to platinum-based chemotherapy, as previously reported in the literature [6,7,8,23,38].

Finally, the multigene panel analysis performed in samples without evidence of BRCA deficiency allowed the identification of a patient with inherited ovarian cancer predisposition due to a *RAD51C* germline variant, with clinical and familial implications, which support the appropriateness of extending genetic testing to clinically meaningful genes other than *BRCA1/2* in OC patients [23].

The main limitation of the study is the small sample size, further reduced in specific sub-analyses due to the unavailability of suitable material for a fraction of cases, which impairs the solidity and statistical significance of figures obtained, though providing further support to existing evidence. However, the comprehensive assessment performed using combined molecular approaches allowed the in-depth characterization of BRCA status and associated genomic features and the provision of meaningful insights into the use and interpretation of predictive BRCA testing.

## 5. Conclusions

The assessment of BRCA status in OC patients provides meaningful prognostic and predictive information. However, analysis of Formalin-Fixed-Paraffin-Embedded samples is prone to false results if not properly developed and validated. Moreover, the implementation of strategies able to detect also epigenetic changes and alterations of other cancer genes may improve the diagnosis of cancers defective for homologous recombination repair mechanisms.

## Figures and Tables

**Figure 1 cancers-15-01530-f001:**
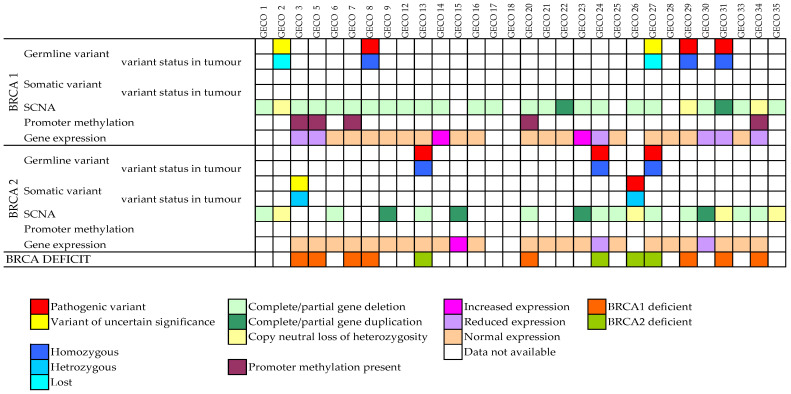
*BRCA1/2* characterization in OC samples. SCNA = somatic copy number alterations.

**Figure 2 cancers-15-01530-f002:**
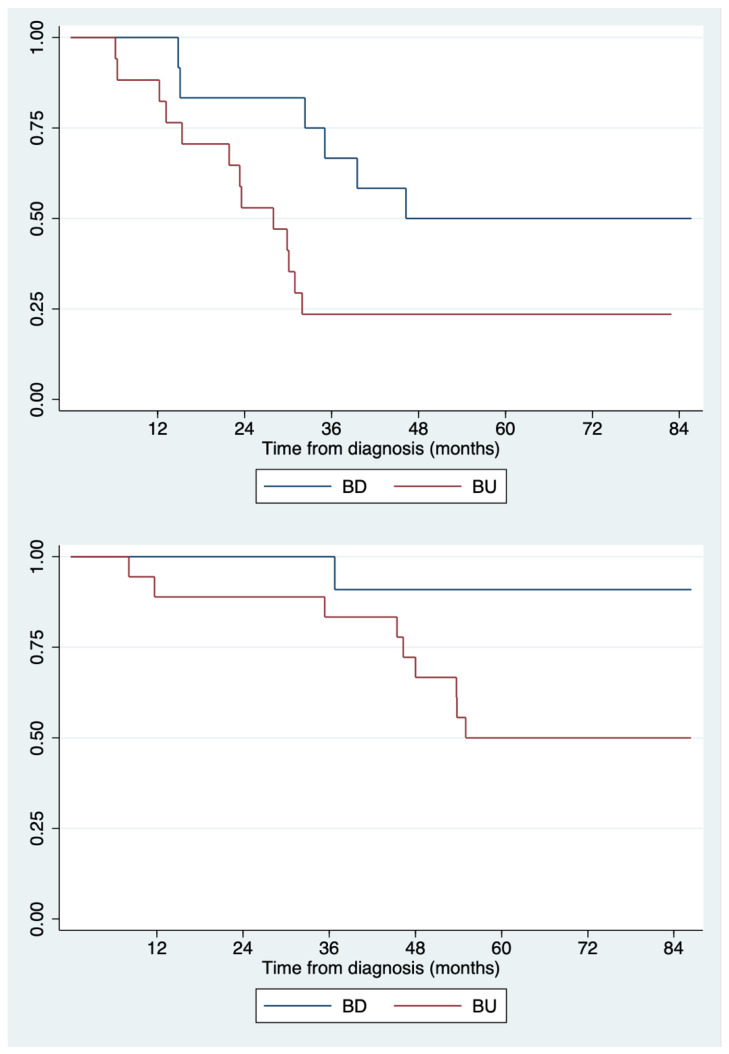
Kaplan–Meier curves illustrating PFS (above) and OS (below) in BD and BU patients. PFS = Progression Free Survival; OS = Overall Survival; BD = BRCA-deficient; BU = BRCA deficit undetected/unclear.

**Table 1 cancers-15-01530-t001:** *BRCA1/2* sequence variants identified.

Gene	Nucleotide Variant	Effect on Protein	Class	Variant Load	Origin	Patient
*BRCA1*	c.547 + 2T > A	?	C4	100%	Germline	GECO 31
	c.569C > T	p.Thr190Ile	C3	20%	Germline	GECO 27
	c.3613G > A	p.Gly1205Arg	C3	25%	Germline	GECO 2
	c.4065_4068delTCAA	p.Asn1355Lysfs	C5	97%	Germline	GECO 29
	c.5123C > A	p.Ala1708Glu	C5	80%	Germline	GECO 8
*BRCA2*	c.1813delA	p.Gly602 = fs*11	C5	90%	Germline	GECO 24
	c.5352C > A	p.Asn1784Lys	C3	50%	Somatic	GECO 3
	c.6442delT	p.Ser2148Leufs*20	C5	60%	Somatic	GECO 26
	c.7558C > T	p.Arg2520Ter	C5	80%	Germline	GECO 13
	c.9118-1G > A	?	C4	80%	Germline	GECO 27

**Table 2 cancers-15-01530-t002:** Array-CGH+SNP analysis results. The table shows CNV features evaluated and compared between tumors of patients with defective and intact patients. Statistically significant values are in bold.

	Patient ID	Total CNVs						Simple CNVs				Complex CNVs			
		n. of CNVs	Unbalanced Genome (%)	Deletions		Duplications		Deletions		Duplications		Deletions		Duplications	
				n.	Average Size (kb)	n.	Average Size (kb)	n.	Average Size (kb)	n.	Average Size (kb)	n.	Average Size (kb)	n.	Average Size (kb)
BU	GECO 6	99	38.56	76	11,736	23	10,664	51	4282	22	10,338	25	26,943	1	17,843
	GECO 9	48	54.00	35	40,948	13	12,270	16	26,677	7	14,230	19	52,967	6	9983
	GECO 16	49	33.78	7	39,263	42	17,177	5	10,600	9	2585	2	33,621	33	49,266
	GECO 18	15	1.59	14	3198	1	2246	13	2349	1	2246	1	14,227	0	/
	GECO 30	37	60.47	33	52,287	4	14,484	4	19,087	2	11,692	29	56,867	2	17,276
	GECO 35	60	54.85	42	27,019	18	26,829	14	4247	4	5180	28	38,405	14	33,105
	Average	51	40.54	35	29,075	17	13,945	17	11,207	8	7712	17	37,172	9	25,495
BD	GECO 27	196	57.74	90	10,995	106	6730	61	3285	87	4882	29	27,210	19	15,193
	GECO 29	72	36.78	42	16,868	30	12,539	41	16,805	20	7607	1	19,453	10	22,403
	GECO 31	90	56.44	37	24,489	53	14,312	29	20,915	34	10,566	8	37,446	19	21,016
	GECO 7	205	43.64	89	4092	116	7956	34	1642	57	2910	55	5606	59	12,832
	GECO 26	107	63.01	47	16,537	60	18,017	18	3690	36	7171	29	24,511	24	34,569
	GECO 13	89	61.26	32	16,157	57	22,624	15	7001	23	5304	17	24,235	34	34,340
	GECO 8	78	44.00	49	15,201	29	19,068	22	1944	19	9775	27	26,003	10	36,725
	Average	120	51.84	55	14,906	64	14,464	31	7897	39	6888	24	23,495	25	25,297
K-S test *p* values		0.015	0.528	0.338	0.091	0.015	0.925	0.068	0.528	0.015	0.712	0.980	0.212	0.245	0.838

**Table 3 cancers-15-01530-t003:** Comparison between sequencing analysis results performed on different types of tissues.

Patient	Gene	Nucleotide Variant	Effect on Protein	Class ^#^	Germline	Variant Load		
						SF Tissues	R-FFPE (2017)	D-FFPE (2020)
GECO 1	*BRCA2*	c.7171G > A	p.Glu2391Lys	C3	N	N	23%	N
GECO 2	*BRCA1*	c.3613G > A	p.Gly1205Arg	C3	Y	25%	20%	13%
GECO 3	*BRCA2*	c.5352C > A	p.Asn1784Lys	C3	N	50%	N	N
GECO 8	*BRCA1*	c.4411G > A	p.Gly1471Ser	C3	N	N	22%	N
	*BRCA1*	c.5123C > A	p.Ala1708Glu	C5	Y	80%	97%	76%
	*BRCA2*	c.5321C > T	p.Pro1774Leu	C3	N	N	28%	N
	*BRCA2*	c.5692G > A	p.Asp1898Asn	C3	N	N	23%	N
GECO 13	*BRCA2*	c.6455C > T	p.Ser2152Phe	C3	N	N	20%	n.a.*
	*BRCA2*	c.7558C > T	p.Arg2520Ter	C5	Y	80%	93%	n.a.*
GECO 17	*BRCA1*	c. 4669G > A	p.Asp1557Asn	C3	N	N	29%	N
GECO 22	*BRCA2*	c.8970G > A	p.Trp2990Ter	C5	N	N	20%	N
	*BRCA2*	c.9968C > A	p.Thr3323Asn	C3	N	N	27%	N
GECO 23	*BRCA1*	c.3679C > T	p.Gln1227Ter	C5	N	N	44%	N
	*BRCA1*	c.5298C > A	p.Ile1766=	C3	N	N	24%	N
	*BRCA2*	c.200G > T	p.Arg67Met	C3	N	N	20%	N
	*BRCA2*	c.2931G > A	p.Leu977=	C3	N	N	33%	N
GECO 24	*BRCA2*	c.1813delA	p.Gly602 = fs*11	C5	Y	90%	100%	n.a. *
GECO 25	*BRCA1*	c.223G > A	p.Glu75Lys	C3	N	N	25%	N
	*BRCA1*	c.1108G > A	p.Val370Ile	C3	N	N	28%	N
	*BRCA1*	c.1411C > T	p.Leu471Phe	C3	N	N	25%	N
	*BRCA2*	c.1621G > A	p.Glu541Lys	C3	N	N	34%	N
	*BRCA2*	c.2420T > C	p.Val807Ala	C3	N	N	42%	N
	*BRCA2*	c.2842G > A	p.Val948Ile	C3	N	N	20%	N
	*BRCA2*	c.3664G > A	p.Ala1222Thr	C3	N	N	27%	N
	*BRCA2*	c.5134G > A	p.Gly1712Arg	C3	N	N	25%	N
	*BRCA2*	c.6712G > A	p.Asp2238Asn	C3	N	N	39%	N
	*BRCA2*	c.8598C > T	p.Phe2866=	C3	N	N	23%	N
	*BRCA2*	c.9562G > A	p.Asp3188Asn	C3	N	N	35%	N
GECO 26	*BRCA1*	c.31G > A	p.Val11Ile	C3	N	N	31%	N
	*BRCA1*	c.3391G > C	p.Asp1131His	C3	N	N	28%	N
	*BRCA1*	c.5227G > A	p.Gly1743Arg	C3	N	N	27%	N
	*BRCA2*	c.484G > A	p.Gly162Arg	C3	N	N	28%	N
	*BRCA2*	c.2967C > T	p.Tyr989=	C3	N	N	28%	N
	*BRCA2*	c.3718C > T	p.Leu1240=	C3	N	N	24%	N
	*BRCA2*	c.3599G > A	p.Cys1200Tyr	C3	N	N	27%	N
	*BRCA2*	c.6158C > T	p.Ser2053Phe	C3	N	N	33%	N
	*BRCA2*	c.6442delT	p.Ser2148Leufs*20	C5	N	60%	88%	92%
GECO 29	*BRCA1*	c.4065_4068del	p.Asn1355Lysfs	C5	Y	97%	97%	n.a. *
GECO 31	*BRCA1*	c.547 + 2T > A	?	C4	Y	98%	97%	n.a. *
GECO 33	*BRCA1*	c.3349G > A	p.Val1117Ile	C3	N	N	26%	N
	*BRCA2*	c.576G > A	p.Met192Ile	C3	N	N	46%	N

C3–5 variants with allelic load >20% are reported. ^#^ pathogenicity class * samples with clear and consistent evidence of pathogenic mutations at SF and R-FFPE were not re-analyzed.

**Table 4 cancers-15-01530-t004:** Clinical characteristics and outcome of patients enrolled.

Patient ID	Age at Diagnosis (Years)	Histopathologic Diagnosis	Site	Grade	Stage ^a^	BRCA Status	Gene Altered in the Germline	Type of Surgery	Post-Surgery Complications	Macro Residual Post- Surgery ^b^	Type of Chemotherapy	Relapse	PARP-i Maintenance Therapy at Relapse	Clinical Status	Time to Relapse (Months)	Time to Death (Months)	Follow-Up Time (Months)
GECO 1	51	PS	FaT + Per	3	IIIc	BU	/	H + BSO	Y	R0	Car + Pac	Y	MD	T	23	/	62
GECO 2	67	PS	Per	3	IIIc	BU	/	H + BSO	Y	R0	NC	Y	N	D	0	11	11
GECO 3	53	PS	Ov	3	IIIc	BD	/	H + BSO	Y	R0	Car + Pac	Y	MD	T	32	/	39
GECO 5	57	PS	Ov	3	IIIc	BD	/	H + BSO	N	R0	Car + Pac	N	/	FU	/	/	84
GECO 6	55	PS	Ov	3	IIIc	BU	/	H + BSO	N	R0	Car + Pac	Y	Y	T	6	/	83
GECO 7	31	PS	Per	3	IIb	BD	/	H + BSO	N	R0	Car + Pac	N	/	FU	/	/	72
GECO 8	33	PS	FaT	3	IV	BD	*BRCA1*	H + BSO	N	R2	Car + Pac	Y	Y	T	15	/	81
GECO 9	50	PS	Ov	3	IV	BU	/	H + BSO	N	R0	Car + Pac	Y	N	D	21	54	54
GECO 12	73	PS	Ov	3	IIIc	BU	/	H + BSO	N	R0	Car + Pac	Y	N	D	27	53	53
GECO 13	65	En	Per	3	IIIc	BD	*BRCA2*	H + BSO	Y	R1	MD	N	/	FU	/	/	82
GECO 14	50	PS	Ov	3	IIIc	BU	*RAD51C*	H + BSO	Y	R0	Car + Pac	N	/	FU	/	/	82
GECO 15	65	PS	Ov	3	IIIa	BU	/	H + BSO	N	R0	MD	N	/	FU	/	/	81
GECO 16	64	PS	Ov	3	IIIc	BU	/	H + BSO	N	R0	Car	Y	Y	D	12	48	48
GECO 17	51	PS	Ov	3	IV	BU	/	H + BSO	N	R0	Car + Pac	Y	Y	D	30	53	53
GECO 18	61	PS	Ov	3	IIIc	BU	/	H + BSO	N	R0	Car + Pac	Y	N	D	23	33	33
GECO 20	39	PS	Ov	3	IV	BD	/	H + BSO	N	R0	Car	Y	Y	T	46	/	71
GECO 21	68	PS	FaT	3	IIc	BU	/	H + BSO	N	R0	Car + Pac	N	/	FU	/	/	80
GECO 22	58	PS	Ov	3	IV	BU	/	H + BSO	Y	R2	Car + Pac	Y	MD	D	15	46	46
GECO 23	48	PS	Ov	3	IIIc	BU	/	H + BSO	Y	R0	Car + Pac	Y	MD	T	32	/	78
GECO 24	67	PS	FaT	3	IIIc	BD	*BRCA2*	H + BSO	Y	R0	MD	N	/	FU	/	/	78
GECO 25	58	PS	Ov	3	IV	BU	/	H + BSO	Y	R0	MD	Y	MD	T	29	/	78
GECO 26	60	PS	FaT	3	IIIc	BD	/	H + BSO	N	R1	Car + Pac	N	/	FU	/	/	78
GECO 27	68	PS	Ov	3	Ic	BD	*BRCA2*	H + BSO	N	R0	Car + Pac	N	/	FU	/	/	79
GECO 28	54	PS	Ov	3	IIIc	BU	/	H + BSO	N	R0	Car + Pac	N	/	T	/	/	69
GECO 29	44	PS	Ov	3	IIIc	BD	*BRCA1*	H + BSO	Y	R0	Car + Pac	N	/	FU	/	/	66
GECO 30	67	PS	Ov	3	IIIc	BU	/	H + BSO	N	R0	MD	Y	N	D	6	8	8
GECO 31	44	PS	Ov	3	IV	BD	*BRCA1*	H + BSO	N	R0	Car + Pac	Y	N	D	15	35	35
GECO 33	65	PS	Ov	3	IIb	BU	/	H + BSO	N	R0	Car + Pac	N	/	FU	/	/	59
GECO 34	49	PS	Ov + FaT + Per	3	IV	BD	/	H + BSO	N	R2	Car + Pac	Y	MD	T	35	/	35
GECO 35	63	PS	Ov	3	IIIc	BU	/	H + BSO	N	R0	Car + Pac	Y	N	D	13	34	34

^a^ FIGO classification. ^b^ Residual tumor (R) classification. PS = Papillary-Serous; En = Endometrioid; BU = BRCA undetected/unclear; BD = BRCA deficiency; H + BSO = Hysterectomy with Bilateral Salpingo-Oophorectomy; Y = Yes; N = No; Car = Carboplatin; Pac = Paclitaxel; NC = No Chemotherapy; MD = Missing Data; PARP-i = PARP-inhibitors; T = in therapy; D = Dead; FU = in Follow-up.

**Table 5 cancers-15-01530-t005:** Comparison of clinical characteristics and outcomes between BD and BU patients.

		BD	BU	*p*
Age at diagnosis, n. (%)	<60 y	8 (66.7)	9 (52.9)	0.703
	≥60 y	4 (33.3)	8 (47.1)	
Histopathologic diagnosis, n (%)	PS	11 (91.7)	17 (100)	0.414
	EN	1 (8.3)	0	
Stage ^a^, n (%)	I-II	2 (16.7)	2 (11.8)	1.000
	III-IV	10 (83.3)	15 (88.2)	
Macroresidual post-surgery ^b^, n (%)	R0	8 (66.7)	16 (94.1)	0.130
	R > 0	4 (33.3)	1 (5.9)	
Relative dose intensity of chemotherapy ^c^, n (%)	>85%	10 (83.4)	14 (82.4)	0.945
	NA	2(16.6)	3 (17.6)	
PFS (mean ± SD, months)		54.9 ± 27.2	34.6 ± 26.7	0.055
		HR 0.382 [95% CI 0.142–1.022]		
OS (mean ± SD, months)		69.1 ± 20.0	58.4 ± 23.5	0.077
		HR 0.155 [95% CI 0.020–1.222]		

PS = Papillary-Serous; EN = Endometrioid; PFS = Progression Free Survival; OS = Overall Survival; BD = BRCA-deficient; BU = BRCA deficit undetected/unclear; HR = Hazard Ratio; NA = Not Available. ^a^ FIGO classification. ^b^ Residual tumor (R) classification. ^c^ Relative dose intensity: ratio of the delivered dose intensity to the standard dose intensity.

## Data Availability

The data presented in this study are available upon request from the corresponding author. The data are not publicly available due to privacy restrictions.

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
