# Peer review of "Characterization of BRCA Deficiency in Ovarian Cancer"

_cancers, 2023, doi:10.3390/cancers15051530_

Round 1

Reviewer 1 Report

It is a good paper with implications in clinical practice

Author Response

We thank reviewer 1 for appreciating our work. We have tried to further improve the paper trying to satisfy the comments of the other reviewers, adjusting the "Simple summary" and "Abstract" parts and adding the "Conclusions" paragraph.

Reviewer 2 Report

I appreciate the opportunity to review this manuscript I looked forward to learning significantly from conclusions, as this is a very active area of research. 

I recommend publication of this manuscript

Author Response

We thank reviewer 2 for appreciating our work. We have tried to further improve the paper trying to satisfy the comments of the other reviewers, adjusting the "Simple summary" and "Abstract" parts and adding the "Conclusions" paragraph.

Reviewer 3 Report

Table 5: It is recommended that authors add a row, detailing the intensity of adjuvant treatment with taxanes and platinum in both groups, since this may be the most important factor that could influence  PFS and OS. 

Author Response

We thank reviewer 3 for reviewing our paper. As suggested, we have added in Table 5 a row detailing the intensity of adjuvant treatment with taxanes and platinum in both groups. Furthermore, we have tried to further improve the paper adjusting the "Simple summary" and "Abstract" parts and adding the "Conclusions" paragraph.